# Niche and Geographic Drivers Shape the Diversity and Composition of Endophytic Bacteria in Salt-Tolerant Peanut

**DOI:** 10.3390/microorganisms13102264

**Published:** 2025-09-26

**Authors:** Xinying Song, Yucheng Chi, Xiaoyuan Chi, Na Chen, Manlin Xu, Xia Zhang, Zhiqing Guo, Kang He, Jing Yu, Ying Li

**Affiliations:** Shandong Peanut Research Institute, National Engineering Research Center for Peanut, Qingdao 266100, China; songxinying88@126.com (X.S.); 87626681@163.com (Y.C.);

**Keywords:** endophytic bacteria, microbial community assembly, saline–alkali soil, peanuts, niche differentiation, geographic conditions

## Abstract

Endophytic bacteria play an important role in the growth, stress tolerance, and metabolic function of salt-tolerant peanuts, yet their community assembly across different saline–alkali soils and plant organs remains poorly characterized. In this study, the V3–V4 variable region of the endophytic bacteria 16S rRNA gene in three organs (roots, leaves, and pods) of high-oleic-acid peanut variety Huayu9118 from three saline–alkali locations (Xinjiang, Jilin, and Shandong, China) was analyzed by high-throughput sequencing. A total of 1,360,313 effective sequences yielded 19,449 amplicon sequence variants (ASVs), with *Proteobacteria* (45.86–84.62%), *Bacteroidota* (6.52–13.90%), and *Actinobacteriota* (3.97–10.87%) dominating all samples. Niche strongly influenced microbial diversity: the roots exhibited the highest level of richness (Chao 1/ACE indices), while the leaves showed the greatest diversity (Shannon/Simpson indices) in XJ samples. Significant compositional differences were observed between aerial (leaves) and underground (roots/pods) organs. Geographic location also markedly shaped endophytic communities, with stronger effects in roots and pods than in leaves—a pattern supported by PCoA combined with ANOSIM (R (roots) = 1, R (pods) = 0.874, R (leaves) = 0.336, respectively, *p* < 0.001). Saline–alkali adaptation led to a marked enrichment of *Novosphingobium* in roots and pods and of *Halomonas* in leaves compared to non-saline–alkali-grown peanuts. Furthermore, the endophytic communities within the same organ type varied significantly across the three saline–alkali sites. Redundancy analysis (RDA) identified the key environmental factors shaping bacterial community composition in the root samples from each location: available phosphorus (AP) and sulfate (SO_4_^2−^) were the strongest predictors in XJ; available potassium (AK) and chloride (Cl^−^) in DY; and hydrolyzed nitrogen (HN), pH, soil organic matter (SOM), and bicarbonate (HCO_3_^−^) in JL. These findings demonstrate that niches and geographical conditions determined the composition and relative abundance of endophytic bacteria in salt-tolerant peanuts, providing new insights into microbial ecological adaptation in saline–alkali ecosystems.

## 1. Introduction

Soil salinization is a land degradation process that results in the excessive accumulation of soluble salts in the soil [1]. In addition to naturally occurring soil salinity, the amount of salinization is also increasing due to irrigation methods and climate change [2,3]. According to the 2024 Global Status of Saline Soil Report by the Food and Agriculture Organization of the United Nations, the total area of saline soil worldwide amounts to 1.381 billion hectares, accounting for 10.7% of the total global land area. China is the third-largest country covered by saline–alkali land in the world [4]. The current area of saline–alkali land in China is approximately 99 million hectares, mainly distributed in the north [5]. Salt-stressed soils affect almost every aspect of plant development, including germination and vegetative and reproductive development, thereby reducing crop yields [6]. In addition, salinity affects soil physicochemical properties and ecological balance [7]. Therefore, soil salinization is an increasingly serious global problem.

Peanut (*Arachis hypogaea*. L) demonstrates considerable potential for cultivation in saline–alkali soils owing to its inherent salt tolerance, as evidenced by germplasm evaluations indicating its suitability as an alternative crop for agricultural restructuring in such marginal environments [8]. Among peanut varieties, the high-oleic-acid (HO) type is particularly noteworthy due to its enhanced nutritional and economic traits. HO peanuts contain a relatively high level of oleic acid, a monounsaturated fatty acid (MUFA), accounting for more than 75% of its total lipid profile, and the ratio of oleic acid to linoleic acid (O/L) is ≥10 [9]. This biochemical profile significantly enhances oxidation stability, extends the shelf life of peanut-derived products, and reduces the risk of rancidity. More importantly, evaluation studies on peanut germplasm have demonstrated that HO peanut varieties such as Huayu 9118 possess notable salt tolerance, and can be promoted as the main varieties for cultivation in saline–alkali land [10]. Thus, the combination of nutritional value, economic benefit, and environmental resilience makes HO peanuts a compelling subject for investigating plant–microbe interactions in stress-prone environments.

Endophytes are non-pathogenic microorganisms that live in the internal tissues of living plants, usually bacteria or fungi [11,12]. Endophytes have a long-term symbiotic relationship with host plants. On the one hand, endophytes promote plant growth and enhance plant stress tolerance and disease resistance through a series of measures such as absorbing water and nutrients [13,14], inducing the production of hormones [15,16], iron carriers [17], and antibacterial secondary metabolites [18]; regulating proline content [19]; and increasing the activity of antioxidant enzymes [20]. On the other hand, plants affect the development process and diversity of endophytic bacteria and fungi through lignification and change the metabolic functions of endophytic bacteria [21], accelerating the colonization of endophytes in host plants [22]. Therefore, the interaction between plants and endophytes is receiving increasing attention.

Endophytes demonstrate significant potential in mitigating both abiotic stresses (e.g., high temperature, drought, and nutrient deficiency) and biotic stresses (such as phytopathogen invasion) [23,24]. However, soil salinity critically impairs microbial viability through dual mechanisms: salt toxicity and reduced water availability, which collectively limit energy substrates for microorganisms [25,26]. This stressor further drives shifts in microbial community structure [27,28]. Notably, salt-tolerant microbes possess unique adaptive traits—including genetic diversity, synthesis of compatible solutes, phytohormone production, and biocontrol capacity—that position them as key agents for salinity resilience. Specifically, plant-growth-promoting rhizobacteria (PGPR) operate through dual modes: some through direct growth stimulation via the provision of fixed nitrogen, phytohormones, siderophore-sequestered iron, and soluble phosphate [29,30], and others indirectly by protecting the plant against soil-borne diseases, most of which are caused by pathogenic fungi [31]. Consequently, research on endophytic communities highlights their strategic value in sustainable agriculture, with empirical evidence confirming that endophyte diversity strongly correlates with host plant health.

Recent studies have provided new insights into the plant microbiome and plant compartment effects using high-throughput sequencing. The distribution of endophytes is influenced by plant genotypes, organs, geographical locations, and growth stages, with significant variations in diversity and composition across these factors [32,33]. For instance, Li et al. [34] observed substantial divergence in endophytic bacterial diversity among plant varieties. Especially in resistant varieties, the community diversity of endophytic bacteria in the root system and the abundance of culturable bacteria are both relatively higher. Similarly, Deng et al. demonstrated that soil parameters—including total carbon, total nitrogen, available phosphorus, and pH—drive compositional differences in root-associated endophytic bacterial communities of *Pennisetum sinese* across five Chinese provinces [35]. Despite these advances, peanut endophyte research remains predominantly reliant on culture-dependent techniques. Papik et al. emphasize that such approaches capture only a fraction of microbial diversity, leaving the full biotechnological potential of endophytes unexplored [17]. Consequently, there are few reports on the research of the community structure and diversity of endophytes in peanuts using high-throughput sequencing. The influence of geographical location and plant ecological niche on the endophytic bacteria of peanuts grown in saline–alkali soil remains elusive.

In this study, we utilized 16S rRNA amplification sequencing technology to evaluate the microbial niche differentiation of endophytic bacterial communities related to the roots, leaves, and pods of multiple saline–alkali land peanuts. By comparing microbial composition and diversity both among plant organs and across geographically distinct saline–alkali and non-saline–alkali sites, we aimed to investigate the influence of plant compartment niche, geographic location, and key soil environmental factors on the structure and assembly of the endophytic microbiota.

## 2. Materials and Methods

### 2.1. Sample Collection and Processing

The root, leaf, and pod samples of healthy HO peanuts (HY9118) were selected from saline–alkali plots in three different areas: Dongying, Shandong Province (37.44° N, 118.68° E); Baicheng, Jilin Province (15.62° N, 122.84° E); and Yili, Xinjiang Uygur Autonomous Region (43.91° N, 81.27° E). Additionally, samples of non-saline–alkali peanuts (HY9118) were taken from Laixi, Shandong Province (36.52° N, 120.31° E). All samples were collected in August 2023. To ensure that the experiment was representative, in each research area, we selected three separate and spatially spaced sampling points. At each sampling point, 10 plants were sampled according to the five-point sampling method. Whole peanut plants were collected using a shovel, with each plant sample placed in a separate sterile plastic bag and then transported to the laboratory on ice. All samples were cut down with sterile scissors, and peanut samples of the same organ from the same sampling point were well mixed to form one composite sample. For each area, there were 3 biological replicates for the different organs. To remove epiphytic microbes, the samples were supplemented with 40 mL of distilled water and two drops of Tween 20 at 25 °C, and then shaken at 220 rpm/min for 20 min. Sequential washing (surface sterilization) was performed with sterile water (20 s), 70% (*v*/*v*) ethanol (30 s), and 2.5% (*v*/*v*) sodium hypochlorite (2 min), followed by rinsing 3–4 times with sterile water. All samples were immediately placed in liquid nitrogen until DNA extraction. Root samples included main and lateral roots. Leaf samples were the third leaf on the main stem of the collected peanut plant (counting from the top downwards). Pod samples were the pods of an entire plant. All sample plots were located in traditional agricultural fields under conventional local farming practices. The three sites have similar farming systems, agronomic management, and fertilization regimes, i.e., according to local recommendations, maintaining a soil moisture level between 50% and 60% through drip irrigation and applying a compound fertilizer of nitrogen, phosphorus, and potassium (N/P/K = 26:15:15) at a rate of 10 kg per acre. All fields adopt the standard cotton–peanut rotation pattern.

### 2.2. DNA Extraction and Quality Testing

The DNA was extracted with the TGuide S96 Magnetic Soil/Stool DNA Kit (Tiangen Biotech (Beijing) Co., Ltd., Beijing, China) according to the manufacturer’s instructions. The DNA concentration of the samples was measured with the Qubit dsDNA HS Assay Kit and Qubit 4.0 Fluorometer (Invitrogen, Thermo Fisher Scientific, Hillsboro, OR, USA).

### 2.3. PCR Amplification and Illumina MiSeq Sequencing

The 335F: 5′-CADACTCCTACGGGAGGC-3′ and 769R: 5′-ATCCTGTTTGMTMCCCVCRC-3′ universal primer pair was used to amplify the V3-V4 region of the 16S rRNA gene from the genomic DNA extracted from each sample. Both the forward and reverse 16S primers were tailed with sample-specific Illumina index sequences to allow for deep sequencing. The PCR was performed in a total reaction volume of 10 μL: DNA template 5–50 ng, Vn F (10 μM) 0.3 μL, Vn R (10 μM) 0.3 μL, KOD FX Neo Buffer 5 μL, dNTP (2 mM each) 2 μL, KOD FX Neo 0.2 μL, and ddH_2_O up to 10 μL. An initial denaturation at 95 °C for 5 min was followed by 25 cycles of denaturation at 95 °C for 30 s, annealing at 50 °C for 30 s, extension at 72 °C for 40 s, and a final step at 72 °C for 7 min. The total obtained PCR amplicons were purified with Agencourt AMPure XP Beads (Beckman Coulter, Indianapolis, IN, USA) and quantified using the Qubit dsDNA HS Assay Kit and Qubit 4.0 Fluorometer (Invitrogen, Thermo Fisher Scientific, Oregon, USA). After the individual quantification step, amplicons were pooled in equal amounts. For the constructed library, an Illumina novaseq 6000 (Illumina, Santiago, CA, USA) was used for sequencing.

### 2.4. Statistical Analysis

The bioinformatics analysis of this study was performed with the aid of the BMK Cloud (Biomarker Technologies Co., Ltd., Beijing, China). According to the quality of a single nucleotide, raw data were primarily filtered by Trimmomatic [36] (version 0.33). The identification and removal of primer sequences were conducted by Cutadapt [37] (version 1.9.1). PE reads obtained from previous steps were assembled by USEARCH [38] (version 10) and followed by chimera removal using UCHIME [39] (version 8.1). The high-quality reads generated from the above steps were used in the following analysis. Clean reads were then processed using feature classification to output ASVs (amplicon sequence variants) using dada2 [37], and the ASVs with an abundance < 0.005% were filtered. The Alpha diversity was calculated and displayed by QIIME2 (version 2020.6.0) and R software, respectively. Beta diversity was determined to evaluate the degree of similarity of microbial communities from different samples using QIIME. Principal coordinate analysis (PCoA) was used to analyze the beta diversity. Furthermore, we employed Linear Discriminant Analysis (LDA) effect size (LEfSe [38]) to test the significant taxonomic difference among groups. A logarithmic LDA score of 4.0 was set as the threshold for discriminative features. The functional predictive analysis of significant enrichment in the peanuts was conducted using FAPROTAX. To explore the dissimilarities of the microbiome among different factors, a redundancy analysis (RDA) was performed in R using the package ‘vegan’. One-way analysis of variance (ANOVA) and Tukey’s HSD test were used to compare the relative abundance of bacteria and the α-diversity index among the different genotypes (varieties) and different habitats (ecological niches) (*p* < 0.05). The relative abundances of the most abundant bacterial taxa in each plant chamber were also compared using one-way ANOVA and Tukey’s HSD test (*p* < 0.05). All statistical analyses were conducted using SPSS version 20.0.

## 3. Results

### 3.1. Statistics of Endophyte Sequences in Peanut Samples

Following quality control and chimera sequence removal, a total of 1,360,313 high-quality bacterial 16S rRNA gene sequences were obtained from 27 peanut samples (Figure 1A). Clustering these sequences at a 97% identity threshold yielded 19,449 amplicon sequence variants (ASVs). Rarefaction curves for all tissue types (leaves, pods, roots) across the three sampling locations (XJ, JL, DY) approached plateaus with increasing sequencing depth (Figure 1B), indicating that the sequencing effort sufficiently captured the majority of bacterial diversity within these libraries. Comparison against the SILVA database revealed distinct ASV compositions among the different tissues. Roots harbored the highest number of unique ASVs (9661), followed by pods (6409) and leaves (5416) (Appendix A). This tissue-specific distribution pattern suggested that the peanut endophytic bacterial communities were strongly influenced by both environmental conditions and agricultural practices. An analysis of the entire dataset (19,449 ASVs) using a petal plot identified 15 core microbial phyla consistently present across all samples (Figure 1C). *Proteobacteria*, *Bacteroidota*, *Patescibacteria*, *Actinobacteriota*, and *Myxococcota* were the predominant phyla. Further taxonomic classification of these ASVs revealed a conserved core microbiome at finer taxonomic resolutions: 28 core classes, 66 core orders, 96 core families, and 104 core genera were identified (Figure 1C), indicating the relative stability and importance of these taxa within the peanut endosphere.

### 3.2. Endophytic Community Diversity and Composition

The alpha diversity analysis of the peanut endophytic bacteria revealed distinct patterns across tissues and regions (Table 1). Consistent across all three regions (XJ, JL, DY), root tissues exhibited the highest microbial richness, as indicated by significantly greater Chao1 and ACE indices compared to leaves and pods. Furthermore, within XJ and JL samples, richness indices followed a clear tissue-specific gradient: root > pod > leaf. In DY samples, however, the pod and leaf richness indices showed minimal differences. In addition, the species diversity patterns, assessed by Shannon and Simpson indices, displayed regional variation. The highest diversity was observed in pods from the JL samples and leaves from XJ samples. In the DY samples, no significant difference in diversity was detected between roots and pods. Collectively, these results demonstrated that within the same ecosystem type and tissue type, significant differences in both richness and diversity indices occurred, highlighting the influence of multiple environmental factors on the peanut endophytic bacterial community.

Beta diversity analysis via principal coordinate analysis (PCoA) was performed to assess the overall structural dissimilarity of endophytic communities across the 27 samples (representing three tissues from three regions). PCoA clearly differentiated the microbial communities among the nine sample groups (Figure 2A). A PERMANOVA test confirmed significant differences in endophytic bacterial community composition across these groups (R^2^ = 0.443, *p* = 0.001). The primary axis of variation (PC1), explaining 15.54% of the total compositional differences, was the main driver separating the samples.

Across all of the samples, a total of 19,449 ASVs were classified into 43 phyla, 106 classes, 306 orders, 658 families, and 1567 genera by python2 (matplotlib-v1.5.1). Among these, 9 phyla and 95 genera exhibited a relative abundance ≥ 1.0% in at least one sample, comprising 49 known and 46 unclassified genera. *Proteobacteria*, *Bacteroidota*, and *Actinobacteriota* were the dominant phyla across all samples, accounting for 45.86–84.62%, 6.52–13.90%, and 3.97–10.87% of the total phyla, respectively (Figure 2B). In the root, *Patescibacteria* was more abundant in the XJ sample than in the JL and DY samples; *Bacteroidota* dominated in the JL sample, while *Actinobacteriota* showed the opposite trend (lower in JL). In leaf tissue, the abundance of *Proteobacteria* was significantly lower in XJ than in the JL and DY samples. Interestingly, the abundance of other phylum microorganisms was higher in the XJ samples, especially *Firmicutes*. In pods, *Proteobacteria* was significantly more abundant in XJ than in the JL and DY samples. These results showed that geography affected bacterial community composition. Furthermore, *Firmicutes* was more abundant in leaves than in pods and roots, and *Patescibacteria* was more abundant in roots compared to pods and leaves. These results showed that the host niche impacted on bacterial community composition. Each tissue has distinct characteristics at the genus level (Figure 2C). For example, *Halomonas*, *Aliidiomarina*, and unclassified_*Halomonadaceae* were the predominant genera in the leaf tissues, especially in the XJ samples. In the root tissue, *Flavobacterium*, *Bradyrhizobium*, unclassified_*Microscillaceae*, *Arenimonas*, *Pseudoxanthomonas*, *Ohtaekwangia*, *Niastella*, and *Steroidobacter* were mainly enriched. Genera like *Hydrogenophaga*, *Novosphingobium*, *Devosia*, and the *Allorhizobium*_*Neorhizobium*_*Pararhizobium*_*Rhizobium* group were also prevalent in roots and pods but nearly absent from leaves. The concentration of *Caulobacter*, *Lysobacter*, and *Altererythrobacter* were the highest in pods.

### 3.3. Host Niche Impacts on Peanut Microbial Community Diversity and Composition

To specifically assess the influence of host niche on the peanut microbiome, we analyzed the bacterial communities within the root, leaf, and pod organs of peanut cultivar HY9118 using 16S rRNA gene (V3 + V4 region) sequencing data from the XJ location. The α-diversity (Shannon and Chao1 indices) of the bacterial communities dramatically differed among organs (Figure 3A). Roots exhibited the highest bacterial α-diversity, while leaves showed the lowest.

The community structure was significantly differentiated, which was evident in a heatmap based on the top 20 genera (Figure 3B). Clustering patterns revealed a clear separation between organ types, indicating that host niche strongly impacts the abundance of dominant microbial taxa. To further clarify the possible interactions between the identified endophytic microbes’ dependencies in peanut organ samples, linear discriminant effect size (LEfSe) was used to quantitatively analyze the biomarkers of different organs. This identified 45 significant biomarkers across the organ samples (Figure 3C,D), with distinct distribution patterns. Leaves harbored the highest diversity of differential biomarkers. These were predominantly associated with the phyla *Firmicutes* and Acidobacteriota, the classes Clostridia and *Bacilli*, and the orders *Enterobacterales*, *Bacteroidales*, and *Lactobacillales*, along with several families, genera, and unclassified species. In contrast, the fewest differential microorganisms were found in the roots of peanut, mainly distributed at the class (three), family (four), and genus (five) levels. In pods, differential microorganisms were mainly affiliated with the phylum *Proteobacteria* (specifically the class *Alphaproteobacteria* and orders *Rhizobiales* and *Burkholderiales*), as well as three families and four genera.

Through a difference analysis based on LEfSe, the distribution of endophytic bacterial phylum levels in different organs was further compared. As can be seen from Appendix A, the phylum-level abundance validation further corroborated the niche-specific differences. Leaves displayed significantly higher relative abundances of *Acidobacteriota* and *Firmicutes* compared to roots and pods. Conversely, the abundance of *Proteobacteria* in peanut leaves was the lowest. Collectively, these analyses demonstrated that host niche (root, leaf, pod) significantly shaped the diversity, composition, and structure of the peanut endophytic bacterial community.

To elucidate the functional implications of the observed niche-specific communities, we performed Functional Annotation of Prokaryotic Taxa (FAPROTAX) analysis on the XJ samples (XJ9118R, XJ9118L, XJ9118F). FAPROTAX revealed significantly enriched specific functional groups across all organs, most notably chemoheterotrophy and aerobic chemoheterotrophy (*p* < 0.05; Figure 4A). The comparative analysis of root (XJ9118R) and leaf (XJ9118L) microbiomes highlighted distinct functional potentials. Leaf endophytes exhibited significant functional advantages in fermentation and nitrate reduction, while the intracellular microorganisms in roots have significant advantages in nitrogen fixation and chemoheterotrophy, dark hydrogen oxidation, and aerobic chemoheterotrophy (Figure 4B).

### 3.4. Geographical Effects on the Peanut Microbiome Composition and Diversity

Next, we conducted a comparative analysis of the microbial communities in the roots, leaves, and pods of peanuts grown in three geographically discrete saline–alkali regions (XJ, JL, DY) against those from a non-saline–alkali control site (LX). PCoA combined with ANOSIM revealed significant geographical structuring of bacterial communities (Figure 5A–C). The strength of this location effect varied by organs: it was strongest in the roots (R = 1, *p* < 0.001) and pods (R = 0.874, *p* < 0.001), and comparatively weaker, though still significant, in the leaves (R = 0.336, *p* < 0.001). In contrast to the beta diversity patterns, only leaf-associated bacterial alpha diversity showed a significant influence of geographical location (Figure 5D–F).

Using the species composition circle diagram, the top 10 most abundant genera were analyzed, revealing the distinct geographical signatures within each organ (Figure 6). In the roots and pods, *Novosphingobium* was highly abundant across all three saline–alkali sites, contrasting sharply with non-saline–alkali land (LX), where *Phenylobacterium* was predominant. Notably, the relative abundance of *Bradyrhizobium* was high in JL9118R and DY9118R samples, but markedly reduced in XJ9118R. Strikingly, *Hydrogenophaga* was mainly distributed in XJ9118F. *Hydrogenophaga* constituted 97.5% of the community in XJ9118F, was minimal (2.5%) in JL9118F, and absent in DY9118F and LX9118F. Similarly, three saline–alkali samples (XJ9118L, JL9118L, and DY9118L) shared similar dominant genera (*Halomonas*, unclassified *Halomonadaceae* and *Aliidiomarina*). Conversely, *Ensifer* and *Bacillus* were mainly distributed in non-saline–alkali land (LX9118L). Interestingly, *Methylobacterium*–*Methylorubrum* was the most dominant genus in JL9118L, DY9118L, and LX9118L, but was absent in XJ9118L. Collectively, these analyses demonstrate that geographical location and saline–alkali stress were the major determinants of endophytic microbial community composition in peanut organs, particularly in roots and pods, with distinct dominant genera characterizing each location–tissue combination.

### 3.5. Relationship Between Environmental Factors and Endophytic Community Composition

To assess the influence of key environmental factors (Appendix A) on the structure of peanut endophytic bacterial communities, we performed a Spearman correlation analysis (Figure 7A). This analysis revealed significant correlations between specific factors and the relative abundance of major endophytic taxa. The concentrations of hydrolyzed nitrogen (HN), pH, soil organic matter (SOM), and bicarbonate (HCO_3_^−^) were significantly positively correlated with the abundance of *Flavobacterium* and unclassified_*Microscillaceae*, but negatively correlated with *TM7a*. Similarly, available phosphorus (AP) and sulfate (SO_4_^2−^) showed significant positive correlations with the abundance of *Allorhizobium*–*Neorhizobium*–*Pararhizobium*–*Rhizobium*, unclassified-*LWQ8* and unclassified_*Rhizobiaceae*. Conversely, they were negatively correlated with the abundance of *Bradyrhizobium* and *Devosia*. In addition, available potassium (AK) and chloride (Cl^−^) were significantly positively correlated with the abundance of *TM7a*, but negatively correlated with *Devosia* abundance. Notably, the abundance of *Bradyrhizobium* exhibited a unique pattern: positively correlated with HN but negatively correlated with AP and SO_4_^2−^. This pattern contrasts sharply with the responses observed for other rhizobial groups (the *Allorhizobium*–*Neorhizobium*–*Pararhizobium*–*Rhizobium* group and unclassified_*Rhizobiaceae*), which showed positive correlations with AP and SO_4_^2−^. A redundancy analysis (RDA) identified the key environmental factors shaping bacterial community composition in the root samples from each location. RDA revealed that in sample XJ9118R, AP (R^2^ = 0.96, *p* < 0.01) and SO_4_^2−^(R^2^ = 0.98, *p* < 0.01) were the strongest predictors. For sample DY9118R, AK (R^2^ = 0.96, *p* < 0.01) and Cl^−^ (R^2^ = 0.95, *p* < 0.01) emerged as the primary influencing factors. In contrast, the bacterial community in JL9118R was most strongly associated with HN (R^2^ = 0.97, *p* < 0.05), pH (R^2^ = 0.95, *p* < 0.05), SOM (R^2^ = 0.95, *p* < 0.05), and HCO_3_^−^ (R^2^ = 0.97, *p* < 0.01) (Figure 7B).

## 4. Discussion

Endophytic bacteria represent crucial microbial resources that engage in dynamic interactions with their plant hosts. Our findings demonstrate significant compositional and diversity differences in the bacterial communities inhabiting distinct peanut organs (roots, leaves, and pods), with notable contrasts also observed between aboveground (leaves) and belowground (roots, pods) organs. Furthermore, the influence of geographical location varied substantially across these different plant compartments.

In recent years, culture-independent high-throughput sequencing has greatly expanded the repertoire of microorganisms known to reside in and on plants as well as in the surrounding environment. However, the research on the endophytes of peanuts mainly relies on culture-dependent isolation and identification. This methodological limitation is reflected in the current literature: for example, Hossain et al. isolated 87 indigenous endophytes from the nodules of peanut roots, characterizing them through molecular, biochemical, and physiological analyses [20]. Similarly, Islam and Mandal isolated 20 endophytic *Actinobacteria* from various peanut tissues, identifying *Micromonospora* sp. ANENR4 as a root colonizer capable of promoting plant growth [40]. Consequently, our application of high-throughput sequencing provides a more comprehensive and cultivation-independent assessment of the peanut endophytic community structure and diversity. Additionally, the research on peanut endophytes mainly focuses on single organs, such as seeds (bacteria) [41], roots (fungi, actinobacteria) [40,42], or specific functional groups like *Rhizobia* (non-nodulating bacteria) [43]. Crucially, comparative analyses across multiple organs or distinctions between above- and belowground compartments have been largely lacking. Our study addresses this gap by systematically comparing the community composition and diversity of endophytic bacteria across different peanut organs and geographical locations.

Regarding the research on the specific endophytic bacteria of salt-tolerant peanut under saline–alkali stress, we found that in the roots and pods, *Novosphingobium* was identified as a dominant endophytic genus. This finding is consistent with its reported role in plant stress adaptation. For instance, Krishnan et al. demonstrated that *Novosphingobium pokkalii* sp. nov not only promoted plant growth through the production of indole acetic acid (IAA), acetoin, and siderophores, but also exhibited strong biofilm formation and the ability to utilize diverse plant-derived organic compounds. Notably, this strain successfully colonized pokkali rice roots even under seawater-level salinity, highlighting its potential for enhancing salt tolerance in host plants [44]. Supporting its role under saline conditions, Wang et al. also proved that *Novosphingobium* was a key biomarker in salt-stressed soybean, further underscoring its importance in saline–alkali adaptation [45]. Similarly, members of the genus *Halomonas* and unclassified_*Halomonadaceae* were dominant in the leaf. *Halomonas* spp. function as effective plant growth promoters under salt stress, attributed to their 1-Aminocyclopropane-1-carboxylic acid (ACC) deaminase activity, reactive oxygen species (ROS)-scavenging capacity [46], and multifaceted plant-growth-promoting (PGP) traits including phosphate solubilization, nitrogen fixation, IAA production, and saline–alkali remediation (e.g., *Halomonas* sp. Bachu 26) [47]. Their dominance in leaves suggests a specialized role in mitigating aerial tissue stressors (e.g., UV radiation, osmotic fluctuations) common in peanut agroecosystems. Collectively, the enrichment of *Novosphingobium* in belowground organs and *Halomonas* in leaves indicates a compartment-specific adaptation strategy within the peanut holobiont, enhancing the host’s tolerance.

In this study, we revealed distinct patterns in microbial diversity across peanut tissues: root communities exhibited the highest richness (Chao1 and ACE indices), whereas leaves from the XJ samples showed the greatest species diversity (Shannon and Simpson indices). This might be due to the different colonization of various microbial inoculation pools habitats: leaves interface with airborne inocula (e.g., dust, rain, aerosols) and phyllosphere-specific stressors (UV radiation, diurnal temperature fluctuations), fostering diverse but transient microbial assemblages. Roots and pods, embedded in soil, face stronger abiotic stress (e.g., pH, moisture gradients) and host-mediated selection (e.g., root exudates), favoring niche-adapted taxa with lower evenness. Meanwhile, the daytime sunshine duration in the Xinjiang region is long, and the intensity of ultraviolet rays is significantly higher than that in other places, resulting in a higher diversity of the aboveground part of the samples in this area than the underground part. The influence of plant organ niches on the composition of microbial communities indicates that the differentiation in microbial communities is driven by biological (plant selection) or abiotic (environmental) factors. The selection of microbial members within the ecological niche of organs may be jointly regulated by various abiotic factors such as the interaction of plant biochemical products, the symbiotic relationship with microorganisms promoting plant growth, and the nutrient and light energy utilization rates of aboveground and underground organs [48].

Our results demonstrate that field location significantly shaped bacterial community composition across all plant compartments, primarily driven by distinct soil properties among the three regions. Xinjiang (Yili) is an inland basin with intense evaporation, and its sulfate-rich strata result in sodium-sulfate-dominated soils. Jilin (Baicheng) is a typical area of soda saline–alkali soil in the Songnen Plain. It is mainly composed of sodium carbonate, so it has extremely strong alkalinity and a high sodium adsorption ratio, with a pH > 9. Shandong (Dongying) is located in the Yellow River Delta. Affected by seawater intrusion and sedimentation, sodium chloride is dominant and the degree of alkalization relatively low. These soil divergences parallel documented location effects in other plant systems, including soybean, *Gymnadenia conopsea*, and *Populus* [33,48,49], suggesting a consistent biogeographical pattern in plant–microbe associations. This intense site effect may reflect the influence of different soil physical and chemical characteristics and nutrients, or other environmental factors such as available moisture, temperature, ultraviolet radiation, and intensity and duration of sunlight on the microbial community [50,51]. In addition to these possible biogeographical explanations, environmental heterogeneity may indirectly affect the microbial community structure in different plant organs through the plasticity of plant functional traits that shape the microbial community.

The Spearman correlation analysis and RDA revealed the preference differences of endophytic bacteria in peanuts under different environmental conditions (Figure 7). The positive correlation between *Flavobacterium*/*Microscillaceae* and HN/SOM is consistent with their known roles in the decomposition of organic matter [52], indicating that these bacterial communities thrive in nutrient-rich rhizosphere environments. In contrast, the negative correlation between *TM7a* and HN/HCO_3_^−^ may reflect its oligotrophic lifestyle in low-nutrient environments. These patterns demonstrated soil chemical properties shaping the formation process of the community. Furthermore, the differences in the responses of various types of *Rhizobia* to AP/SO_4_^2−^ highlight their functional specialization: the positive correlation of the “*Allorhizobium*–*Neorhizobium*–*Rhizobium*” complex with AP indicates that it has a higher demand for phosphorus in terms of nodule formation efficiency [53], while the negative phosphorus uptake response of *Bradyrhizobium* in high-phosphorus soil may indicate the presence of competitive exclusion, which is consistent with the known adaptation of this species to low-phosphorus environments [54]. This niche differentiation could explain their spatial segregation in agricultural soils with heterogeneous P distribution. These findings indicate that the balanced management of phosphorus and nitrogen is crucial for maintaining a diverse population of nitrogen-fixing bacteria. This provides operational guidance for growing peanuts in different soil conditions.

While this study provides insights into the geographic and compartment-specific patterns of the peanut endophytic microbiome under saline–alkali conditions, several limitations should be considered. First, the use of a single, salt-tolerant peanut variety (Huayu 9118) prevents us from discerning host genotype-dependent microbial associations. Second, our 16S rRNA amplicon sequencing approach reveals community structure, but not functional activity; thus, the ecological roles of the identified taxa remain inferred. Future research should incorporate multiple genotypes under controlled conditions, and employ transcriptomic analyses of these samples to directly quantify the functional activity of the identified microbial communities and elucidate their specific responses to saline–alkaline stress. To further resolve organizational complexity within organs, techniques such as laser-capture microdissection (LCM) should be employed to characterize microbial assemblages at sub-organ resolution—enabling the association of endophytes with specific cell types (e.g., xylem, phloem, or meristem) and illuminating their niche-specific roles in stress adaptation. Moreover, while this study provides a valuable snapshot of community structure under steady-state conditions, the dynamic and reciprocal interactions within the plant holobiont remain largely uncaptured. Time-series experiment sampling across diurnal cycles and discrete stress events (e.g., post-irrigation or heatwaves), integrated with multi-omics approaches such as metatranscriptomics and metabolomics applied to both host and microbiota, will be essential to decipher the rapid, mechanistic dialogues underlying microbial assembly and resilience. Together, these approaches will bridge the gap from correlation to causation, ultimately enabling the targeted design of microbial consortia for enhancing crop tolerance in saline–alkaline environments.

## 5. Conclusions

This study demonstrates that both plant niche and geographic location collectively shape the assembly and composition of the endophytic bacterial communities in salt-tolerant peanuts. Saline–alkali conditions consistently enriched specific beneficial taxa, including *Novosphingobium* in roots and pods and *Halomonas* in leaves, suggesting their potential role in host adaptation to stress. A redundancy analysis identified key soil factors—such as AP, SO_4_^2−^, AK, Cl^−^, HN, pH, SOM, and HCO_3_^−^—as major drivers of microbial variation in the root endosphere. These findings provide new ecological insights into peanut–microbe interactions under saline–alkaline stress.

## Figures and Tables

**Figure 1 microorganisms-13-02264-f001:**
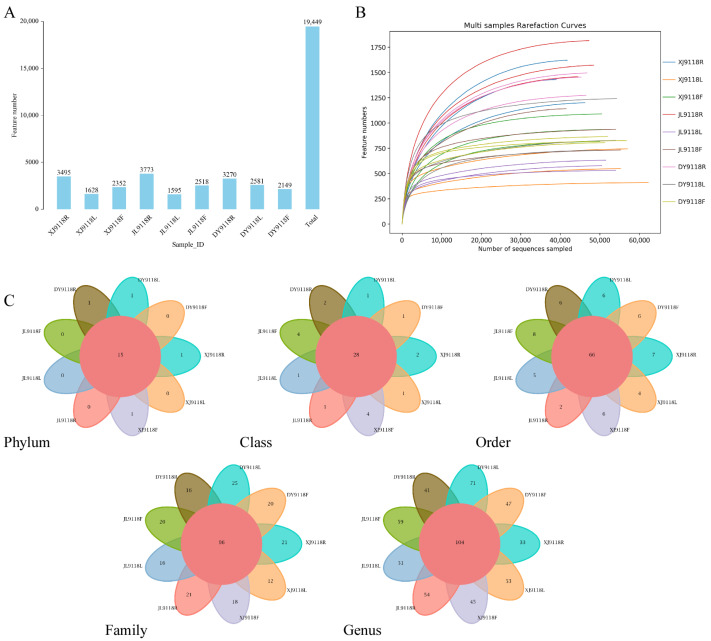
Composition of the endophytic bacterial communities across peanut organs and geographic locations. (**A**) Bar plot comparing the number of amplicon sequence variants (ASVs) identified in pod (F), leaf (L), and root (R) samples from the three saline–alkali regions (Xinjiang, XJ; Jilin, JL; Dongying, DY). (**B**) Rarefaction curves illustrating sequencing depth and sample adequacy across all samples. (**C**) Core microbiota analysis of all samples at the phylum, class, order, family, and genus levels.

**Figure 2 microorganisms-13-02264-f002:**
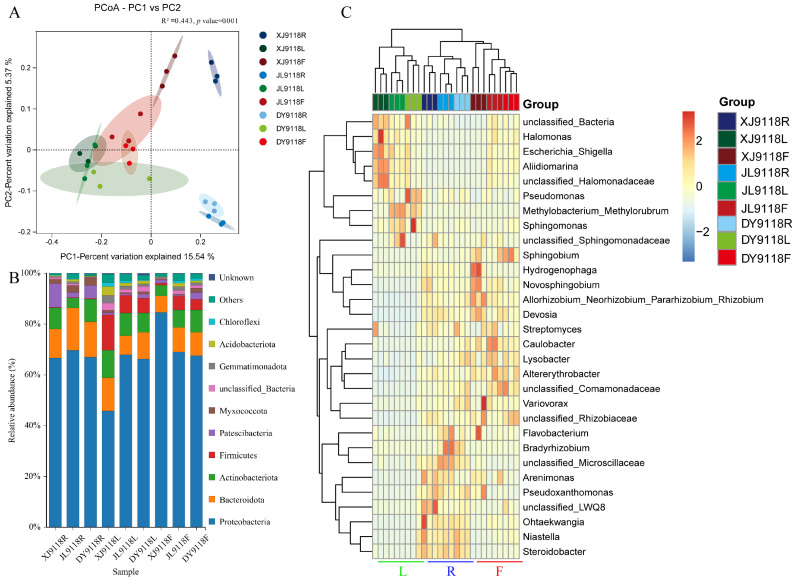
Endophytic community diversity and composition. (**A**) Principal coordinate analysis (PCoA) of endophytic bacteria. The contribution rates of PC1 to the differences in the composition of endophytic bacterial communities was 15.54%. (**B**) Species abundance of microbial communities in different samples at the phylum level (top 10). (**C**) Heat map of clustering distribution of microbial communities in different samples at the genus level (top 20). L: leaf, R: root, F: pod.

**Figure 3 microorganisms-13-02264-f003:**
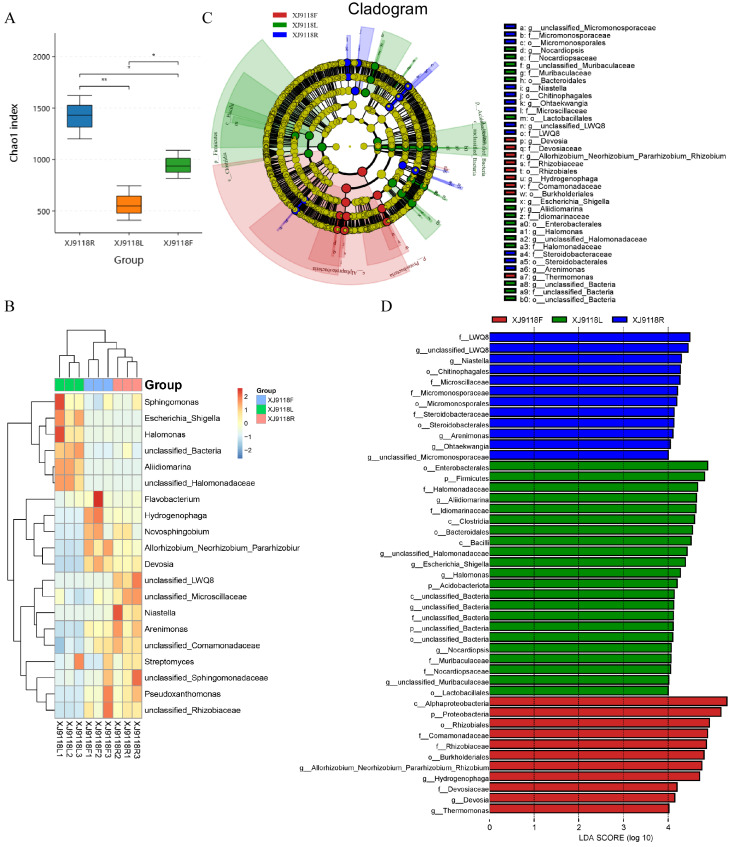
Host niche impacts on peanut microbial community diversity and composition. (**A**) Alpha diversity of microbial communities in different niches. * *p* < 0.05; ** *p* < 0.01. (**B**) Impact of host niche on microbial distribution at the genus level (top 20). (**C**) Cladogram of bacterial taxa with differential abundances among three different niches based on LEfSe software (version 1.1.1) analysis. (**D**) Histogram of LDA value distribution. Note: p (Phylum), c (Class), o (Order), f (Family), and g (Genus).

**Figure 4 microorganisms-13-02264-f004:**
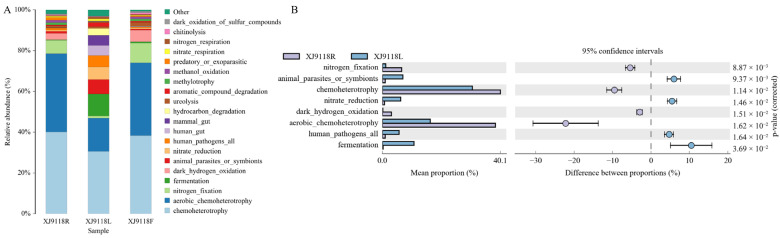
Predicted functional analysis of the peanut microbiome. (**A**) Functional predictive analysis of significant enrichment in peanut using FAPROTAX. (**B**) Analysis diagram of the differences in KEGG metabolic pathways at the second level of microorganisms in roots and leaves. In the figure, the purple color represents sample XJ9118R, and the blue color represents sample XJ9118L. The left side shows the abundance ratios of different functions in the two groups of samples, the middle one shows the difference ratios of functional abundances within the 95% confidence range, and the value on the far right is the *p* value.

**Figure 5 microorganisms-13-02264-f005:**
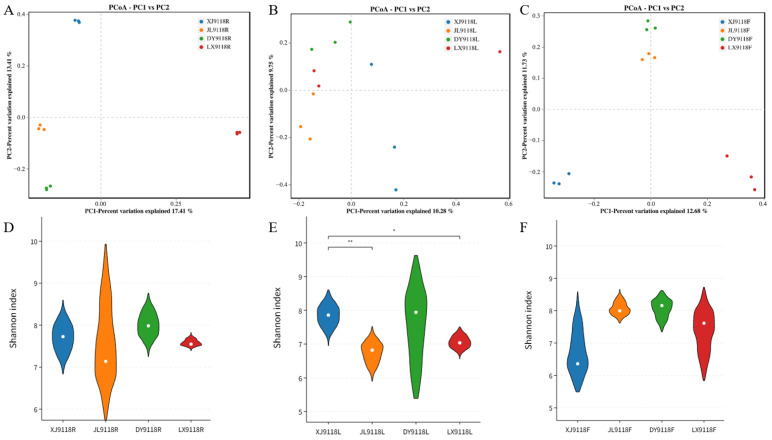
The diversity of the microbial community. (**A**–**C**) The PCoA analysis of bacterial communities based on Bray–Curtis distances. (**D**–**F**) The Shannon index of bacteria. (**A**,**D**) The diversity of the microbial community in the roots of plants from four different geographical locations. (**B**,**E**) The diversity of the microbial community in the leaves of plants from four different geographical locations. (**C**,**F**) The diversity of the microbial community in pods of four different geographical locations. ** refers to significant differences between each site based on the Wilcoxon test (*p* < 0.05); * refers to significant differences between each site based on the Wilcoxon test (*p* < 0.1).

**Figure 6 microorganisms-13-02264-f006:**
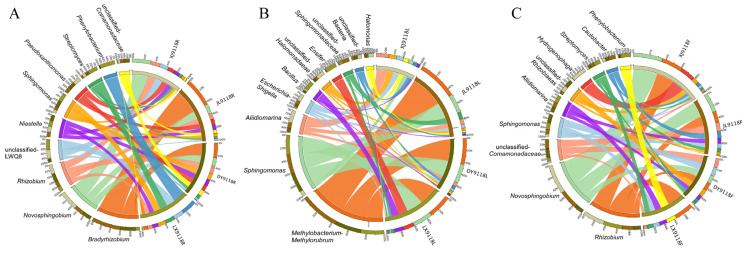
Circus diagram of species composition. (**A**) Circus diagram of species composition in HY9118 roots from four different geographical locations. (**B**) Circus diagram of species composition in HY9118 leaves from four different geographical locations. (**C**) Circus diagram of species composition in HY9118 pods from four different geographical locations. In the Circus diagram, one side represents the sample information and the other side represents the species, represented by different colors. The ribbons in the figure indicate the presence of a certain species in a sample. The width of the color is related to the relative abundance of the species in the sample. The thicker the ribbon, the richer the content of the species.

**Figure 7 microorganisms-13-02264-f007:**
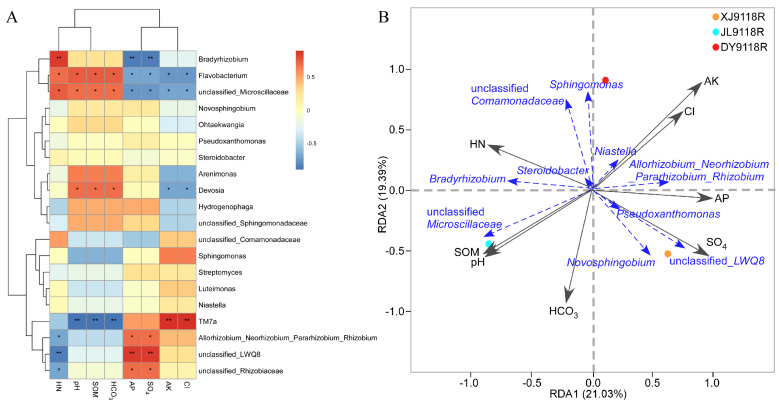
Relationships between dominant bacterial genera and environmental factors across saline–alkali conditions. (**A**) Spearman correlation analysis between the top 20 bacterial genera and environmental factors. * *p* < 0.05; ** *p* < 0.01. (**B**) The redundancy analysis (RDA) identified the key environmental factors shaping bacterial community composition in the root samples from three saline–alkali lands. HN: hydrolyzed nitrogen, SOM: soil organic matter, HCO_3_: bicarbonate, AP: available phosphorus, SO_4_: sulfate, AK: available potassium, and Cl: chloride. Statistical significance was calculated by R (version 3.3.1).

**Table 1 microorganisms-13-02264-t001:** The microbial community richness and diversity indices of pod (F), leaf (L), and root (R) samples collected from three locations (XJ, JL, DY) at a 97% similarity threshold.

Sample	ACE	Shannon	Simpson	Chao1
XJ9118R	1424.11 ± 209.45 a	7.71 ± 0.32 ab	0.98 ± 0.011 abc	1417.71 ± 210.08 a
XJ9118L	569.58 ± 170.38 c	7.85 ± 0.29 a	0.99 ± 0.001 a	567.48 ± 168.05 c
XJ9118F	949.47 ± 136.11 b	6.72 ± 0.63 b	0.96 ± 0.017 bc	947.52 ± 136.28 b
JL9118R	1623.91 ± 178.62 a	7.62 ± 0.86 ab	0.95 ± 0.034 c	1615.42 ± 181.43 a
JL9118L	581.44 ± 50.45 c	6.75 ± 0.29 b	0.97 ± 0.006 abc	579.73 ± 50.07 c
JL9118F	940.20 ± 204.58 b	8.07 ± 0.19 a	0.99 ± 0.003 a	936.91 ± 204.46 b
DY9118R	1415.18 ± 116.02 a	8.01 ± 0.27 a	0.98 ± 0.010 ab	1407.80 ± 117.14 a
DY9118L	936.78 ± 272.96 b	7.57 ± 1.01 ab	0.98 ± 0.012 ab	933.31 ± 271.28 b
DY9118F	835.46 ± 31.86 bc	8.06 ± 0.26 a	0.99 ± 0.005 a	832.27 ± 31.19 bc

Values are means ± standard error (*n* = 3). Statistical significance was calculated using Dunn’s test. The same letter represents no significant difference; different letters represent a significant difference (*p* < 0.05).

## Data Availability

The original contributions presented in this study are included in the article/Appendix A. Further inquiries can be directed to the corresponding author.

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
