# Peer review of "Niche and Geographic Drivers Shape the Diversity and Composition of Endophytic Bacteria in Salt-Tolerant Peanut"

_microorganisms, 2025, doi:10.3390/microorganisms13102264_

Round 1

Reviewer 1 Report (Previous Reviewer 3)

Comments and Suggestions for Authors

The new for of the manuscript present well all the data obtained by the authors, with clear tables and figures.

Overall, the manuscript is interesting and bring new data to the field.

Author Response

Comments1: The new for of the manuscript present well all the data obtained by the authors, with clear tables and figures.

Overall, the manuscript is interesting and bring new data to the field.

Response: We are deeply grateful for your positive assessment of our work and for acknowledging the clarity of the data presentation. We sincerely appreciate the time and effort you dedicated to reviewing our manuscript and providing constructive feedback, which has greatly contributed to improving its quality. Your encouraging comments are a valuable recognition of our efforts to advance understanding in this field.

Thank you once again for your thoughtful and supportive review.

Reviewer 2 Report (New Reviewer)

Comments and Suggestions for Authors

Niche and Geographic Drivers Shape the Diversity and Composition of Endophytic Bacteria in Salt Tolerant Peanut

Abstract

Use as keywords significant words for the study but different from that in title.

Introduction

Lines 52 to 60

Authors must start describing the potential of peanut plants to grow in salty soils and then describing the plant.

See comments in MS.

Materials and methods

In section 2.1 Sample collection and processing

Indicate clearly that sample plots were agricultural traditional fields or well control fields. From the description it is inferred that is the first case. Indicate it.

Results

Consider to include some figures from supplementary material into the main MS. The extension of the main MS is short.

Figure 2 (main MS) can not be read. Consider this comment to improve all figures.

Discussion

Pose the limitations of the study.

Conclusion

Ok

Author Response

This manuscript is a resubmission of an earlier submission. The following is a list of the peer review reports and author responses from that submission.

Round 1

Reviewer 1 Report

Comments and Suggestions for Authors

The manuscript by Song, Chi and colleagues, entitled “Niche and Geographic Drivers Shape the Diversity and Composition of Endophytic Bacteria in Salt-Tolerant Peanut”, describes the endophytic microbiome of a salinity-tolerant peanut variety grown in three salt stress-inducing sites with different soil properties (aridity, alkalinity or saltwater intrusion). The analysis is performed on root, pod and leaf tissues.

The Authors conclude that plant tissue, rather than growing location, is the main driver of bacterial endophytic communities, and provide indications based on predictive tools, concerning the key functions of bacterial taxa.

Although representing a potential interest, the work does not seem fit for publication in the present state, as the experimental design appears flawed in relation to the declared aims of (1) evaluating microbiome composition differences in different tissues, and (2) identifying ecological functions of the microflora that may be exploited in field practice. The experiments included one stress-tolerant peanut variety, with three tissues considered separately, growing in three sites described as stressful. For each tissue/site combination, the collected plant material was pooled, and three replicates were taken from the pool.

In these experimental conditions, there is no way to know whether the microbiome is relevant to stress tolerance, in absence of controls such as susceptible peanut genotypes growing in the same sites, or the same genotype growing in non-stressing conditions. If performing further experiments is not in the Authors’ range of possibilities, they may consider extracting relevant data from previously published literature on peanut microbiomes.

Observed differences among tissues might be accepted as legitimate; however, being drawn from the same batch, experimental replicates cannot be considered independent, invalidating the statistical analysis.

Little information is provided on plant biometric parameters (eg. photosynthetic efficiency, stress metabolites) indicating the stress level, or on the plot management, such as watering, fertilization and crop rotation. These activities may affect both the stress levels suffered by the plant and the microbiome composition.

Any consideration on microbiome functional traits relies on in silico elaboration and prediction, with no further biological testing. Overall, such conclusions appear weak and speculative.

Information provided in fig. 6 would be useful in this sense; however, it still seems dubious, as it is based on a non-parametric test performed on few data points and with no control on the variables. The Authors may consolidate their findings by testing the correlation between the population of selected taxa (assessed eg. by qPCR) and characteristics in experimentally amended soils (eg. JL +2, 5, 10% salt; DY +10, 20 50 mg bicarbonate, ...).

Please find more details in the attached file.

Reviewer 2 Report

Comments and Suggestions for Authors

This research is timely and of high interest to academic and research applying readers from various disciplines and especially for safer food production.

The authors state that the presented 'results indicated that niches and geographical conditions determined the composition and relative abundance of endophytic bacteria. This conclusion is helpful for understanding the relationship and ecological functions of endophytic bacteria in peanuts, and provides valuable information for identifying functional endophytic bacteria, as well as promoting the targeted manipulation of beneficial microorganisms for benefiting high-oleic acid peanuts health in saline-alkali ecosystems.'

While the study is in line with frequently applied views, methods and intended applications, I have important concerns related to the basic understanding of endophytic bacterial functionality and the perspectives derived from the approach.

Specifically, the approach neglects, to my view, that sampling did not/could not consider comparable conditions, means identical day times, actual wheather conditions, actual growth phases and responses on identical momentary stress situations (rain, temperature, wind etc). However, these conditions can shape/shift the whole bacterial community response and therefore, the validity of comparable results. Furthermore, current knowledge points to the importance of quantitative whole bacterial community responses (seen in transcriptoms) rather than the relevance of the quality of present bacterial communities, especially also in response to pathogen attacks as was recently shown for Xylella threats.

Also, the presented analyses refer to organs, namely roots, leaves and pods (later these are referred as 'tissues'). However, endophyte populations may critically differ in their responses depending on the target cells or true tissues of stress/pathogen attacks. This means, there might be relevant differential responses, for example, between meristem, xylem and phloem cells that can be critical for whole plant tolerant behavior. This is not considered in the manuscript.

Recent studies also show that the endophyte bacterial community as a whole can rapidly interact with the dynamic metabolic stress response of the hosting plant (see discussions on functional marker development that need to consider the holobiont nature of plants. However, if samples are taken from various locations without concerning this fine-tuned short time- and specific condition-dependent interplay within the plant holobiont, comparisons between locations and using plant organs as a whole will not be sufficiently valid for the objectives of the study.

In order to be able to publish nevertheless the performed well-done research, the manuscript should be improved by discussing these aspects in the manuscript. I would recommend besides including according references into the introduction do this in the conclusion while discussing about perspectives of further research. Without doing this, the conclusions/perspective might lead other researchers into the wrong direction.

The reviewer (retired volunteer senior/independent scientist) uses importantly own insights from recently published own research based on wide literature studies without the intention to influence or manipulate incorrectly the reviewed manuscript. 

I simply hope that it can be helpful.

Reviewer 3 Report

Comments and Suggestions for Authors

The research on endophytic bacteria is important in the current context of better understanding plant microbiome and its importance in crop success under saline and alkaline ecosystems.

The current form of the manuscript "Niche and Geographic Drivers Shape the Diversity and Composition of Endophytic Bacteria in Salt-Tolerant Peanut" can be improved based on several suggestions.

Abstract - this section can be improved, by reducing the Introduction and Aim parts, along with the last sentence (lines 25-28) and add more of the findings from the study.

The Introduction present all the necessary information to state the importance of the study and how the results will improve the knowledge in the field. The aim is long and can be presented in a condensed form, with the addition of several objectives or hypotheses.

Materials and methods are informative and present the entire flow of the research.

Results section - the authors present multiple data, in detailed tables and graphs, along with multiple statistical analysis, which makes this section explicit. Each test and result is explored and interpreted.

The Discussion section links the trends from the research with other similar international studies, which assure the place of the results within the field.

Conclusion section - I suggest the replacement of first and last sentence from this section with other findings from the manuscript.